# Hot Air Drying of Seabuckthorn (*Hippophae rhamnoides* L.) Berries: Effects of Different Pretreatment Methods on Drying Characteristics and Quality Attributes

**DOI:** 10.3390/foods11223675

**Published:** 2022-11-16

**Authors:** Shudan Tan, Yu Xu, Lichun Zhu, Zhihua Geng, Qian Zhang, Xuhai Yang

**Affiliations:** College of Mechanical and Electrical Engineering, Shihezi University, Shihezi 832003, China

**Keywords:** seabuckthorn berries, pretreatment, hot air drying, drying characteristics, quality attributes, coefficient of variation method

## Abstract

Seabuckthorn berries are difficult to dry because the outermost surface is covered with a dense wax layer, which prevents moisture transfer during the drying process. In this study, uses of ultrasonic-assisted alkali (UA), pricking holes in the skin (PH) and their combination (UA + PH) as pretreatment methods prior to hot air drying and their effects on drying characteristics and quality attributes of seabuckthorn berries were investigated. Selected properties include color, microstructure, rehydration capacity, as well as total flavonoids, phenolics and ascorbic acid contents. Finally, the coefficient of variation method was used for comprehensive evaluation. The results showed that all pretreatment methods increased the drying rate; the combination of ultrasonic-assisted alkali (time, 15 min) and pricking holes (number, 6) (UA15 + PH6) had the highest drying rate that compared with the control group, the drying time was shortened by 33.05%; scanning electron microscopy images revealed that the pretreatment of UA could dissolve the wax layer of seabuckthorn berries, helped to form micropores, which promoted the process of water migration. All the pretreatments reduced the color difference and increased the lightness. The PH3 samples had the highest value of vitamin C content (54.71 mg/100 g), the UA5 and PH1 samples had the highest value of total flavonoid content (11.41 mg/g) and total phenolic content (14.20 mg/g), respectively. Compared to other pretreatment groups, UA15 + PH6 achieved the highest quality comprehensive score (1.013). Results indicate that UA15 + PH6 treatment is the most appropriate pretreatment method for improving the drying characteristics and quality attributes of seabuckthorn berries.

## 1. Introduction

Seabuckthorn (*Hippophae rhamnoides* L.) belongs to the deciduous shrub or small arbor of the genus seabuckthorn in the family Elaeagnaceae. The seabuckthorn berries are 4–10 mm in diameter, 1–2 cm in length, 0.2–1.0 g in weight, orange-yellow or orange-red in color, and spherical or oblong in shape. It is also called vinegar willow and sour thorn due to its sour and astringent taste [1]. In addition, seabuckthorn berries are rich in nutrients and have good bioactive functions, such as antioxidant, anti-tumor, hypolipidemic and hypoglycemic functions [2,3,4].

Freshly picked seabuckthorn berries have high moisture contents of up to 70–85% [5], which makes them very easy to soften and deteriorate after harvesting at room temperature, thus, they are mostly preserved by freezing. Furthermore, the thin skin of seabuckthorn berries is easy to break, which is very unfavorable for long-term preservation and long-distance transportation. Drying is an effective method that can not only reduce the loss of active ingredients during long-term cryopreservation but also reduce storage and transportation costs [6]. At present, the common drying methods of seabuckthorn berries are natural drying and hot air drying (HAD). Lin et al. [7] observed that the drying time of HAD was shortened by 67.93% compared with natural drying. Hence, HAD is suitable for the production and processing of seabuckthorn berries. However, the outermost surface of seabuckthorn berries is covered with a wax layer, which restricts moisture transfer from inside to outside during the drying process, leading to difficulties in drying, including prolonged drying time, oxidative degradation, and serious quality deterioration. Therefore, finding a pretreatment method to destroy the wax layer and decrease drying time has become an indispensable step in seabuckthorn berries processing. Recently, several pretreatment methods have been developed to destroy the wax layer and improve the efficiency and quality of drying, such as chemical dipping, ultrasound, drilling small holes in the skin, halving, etc. [8,9,10,11,12,13]. Yao et al. [14] observed that the samples treated with blanching (70 °C), 2% Na_2_CO_3_, 2% grape dry-promoter and 2% NaHCO_3_ could significantly shorten the drying time of seabuckthorn berries. Among them, the samples treated with 2% Na_2_CO_3_ were the most effective and had the highest quality. However, chemical pretreatment still has limitations, including chemical remains on the fruit surface. Therefore, we should choose food-grade chemicals and wash after pretreatment. Zhang et al. [10] observed that perforation pretreatment could shorten the drying time of peppers. This method does not require the use of chemicals, but it can cause juice loss from the sample. Hence, it is necessary to reduce this loss by decreasing the perforation size. Ultrasonic treatment has been proven to destroy the wax layer of fruits. Furthermore, Gao et al. [15] found that during the ultrasonic-assisted lye (UA) peeling of tomatoes, the ultrasonic treatment and lye treatment had a synergistic effect. This would suggest potential applications in the wax layer destruction of other fruits. Based on the above conclusions, we propose that UA and PH may be efficient pretreatment methods for seabuckthorn berries. Currently, no systematic reports have been found exploring the impact of those methods on berries pretreating.

Accordingly, the objective of this study was to compare the effects of PH, UA and UA + PH pretreatments on drying kinetics and quality attributes of dried seabuckthorn berries, and to select the most suitable pretreatment method and parameter according to the comprehensive scores that calculated by means of the coefficient of variation method, which provides a theoretical basis for the future actual production.

## 2. Materials and Methods

### 2.1. Materials and Chemical Reagents

Frozen seabuckthorn berries, species of late autumn red, were purchased from a seabuckthorn planting base, which were harvested and transported from E’min (Xinjiang, China). The initial moisture content of the thawed seabuckthorn berries was measured using the oven-drying method at 80 °C until achieving a constant weight and the value was determined as 81.91% ± 0.86% in wet basis (w.b.) or 4.53 ± 0.01 *g*/*g* in dry basis (d.b.) [16]. To prevent moisture loss, the fruits were packed in plastic bags and kept frozen in a refrigerator at −(18 ± 1) °C before conducting experiments. For biochemical analysis, such as ascorbic acid, total flavonoid content, total phenolic content, the various chemicals used were purchased from Shihezi North Biological Trading Co., Ltd. (Xinjiang, China). These include sodium carbonate anhydrous, methanol and absolute alcohol, oxalic acid, 2,6-dicholorophenolindophenol. Moreover, standards include L-ascorbic acid, rutin and gallic acid.

### 2.2. Pretreatment Methods

Prior to the experiment, seabuckthorn berries were washed three times with tap water, then dried with tissues. To ensure the uniformity of the experimental results, which could be influenced by the physical characteristics of agricultural products, we selected samples with similar size, color and weight and without any damage to the skin (average diameter, 0.67 ± 0.1 cm; mean length, 1.17 ± 0.08 cm; average weight, 0.5 ± 0.1 g) for the pretreatments and drying experiments. For each method of pretreatment, a 100 ± 1 g sample of seabuckthorn berries was used.

The following pretreatment parameters were selected according to the results of pre-experiments and the research results of related articles [8,17]. If the ultrasonic temperature is higher than 30 °C or the ultrasonic time is more than 15 min, the skin of seabuckthorn berries will be broken and the juice will lose easily. If the ultrasonic power is too low and the time of ultrasound is less than 5 min, the dissolution of the wax layer exhibits no significant effects [12]. If the hole numbers are too many, the juice of samples will lose rapidly. According to the size of seabuckthorn berries, the perforated depth was 1.5 mm, and the piercing positions were evenly distributed at the largest diameter (Figure 1).

The same HAD process was applied to seabuckthorn berries under different treatments:
(i)The control group (CG). Without any pretreatment and dried by HAD were used as the control group. It means that the seabuckthorn berries only need to be frozen for 12 h at −18 °C and quickly transferred to HAD.(ii)Prick holes in the skin (PH). The only difference from the control group was that pricking holes (number, 1, 3, 6; abbreviation, PH1, PH3, PH6) in the skin of frozen seabuckthorn berries quickly using a homemade metal needle with a diameter of 0.9 × 10^−3^ m before HAD.(iii)Ultrasonic-assisted alkali (UA). The seabuckthorn berries were placed in a beaker with 3% Na_2_CO_3_ and introduced into an ultrasonic bath (Ultrasonic Instruments, KQ5200DE, Kunshan, China) for 5, 10, 15 min separately (abbreviation, UA5, UA10, UA15); the frequency and power of ultrasound were 40 kHz and 200 W, respectively. The temperature during UA was noted to be always lower than 30 °C. Then, they were washed three times with tap water and dried with tissues. Finally, they were frozen for 12 h at −18 °C and quickly transferred to HAD.(iv)The combination of ultrasonic-assisted alkali and pricking holes in the skin (UA + PH). UA + PH was carried out for each set of ultrasonic time in combination with each number of pricking holes, therefore making a total of 9 trail sets such as- (UA5 + PH1, UA5 + PH3, UA5 + PH6, UA10 + PH1, UA10 + PH3, UA10 + PH6, UA15 + PH1, UA15 + PH3, UA15 + PH6). This method includes the following steps: finishing the work of UA until the frozen time’s over, pricking holes quickly before HAD.

### 2.3. Microstructure

A Scanning Electron Microscope (SU8000, Hitachi Co., Tokyo, Japan) was used to observe the influences of different treatments made to the microstructure of the surface morphologies. In order to reduce the effects of hot air drying on the skin of seabuckthorn berries, the samples were dried by freeze-drying (1-2 LD plus, Marin Christ Co., Osterode, Germany). The skin of samples was carefully separated and cut with a sharp blade, then fixed on the sample stage with conductive adhesives and sputtered coated with gold, observed at 5.0 KV and 500× magnification.

### 2.4. Hot Air Drying

Pretreated samples were spread in one-layer on stainless steel sieves carpeted with a soft silicone pad to avoid sticking and dried in a hot air dryer (DHG-9070A, Yiheng Technology Co., Ltd., Shanghai, China). Turn on the drying equipment 30 min earlier, set the temperature to 60 °C and air velocity to 2.2 m/s. These parameters were selected and relied on a seabuckthorn berries processing plant and relevant literature, respectively. During the drying process, weighing every 60 min for the first 10 h and every 120 min after, using an electronic balance (BSM-4220.4, Zhuojing Electronic Technology Co., Ltd., Shanghai, China). The weighing time was no more than 10 s at one time. The drying test was stopped when the final wet basis moisture content was lower than 15% (w.b.) or the dry basis (d.b.) moisture content was lower than 0.18 *g*/*g* following the method mentioned in [16] 191. To obtain the exact mean value, all experiments were repeated three times. The dried samples were cooled to room temperature (23 ± 2) °C, packed using a zip-lock polyethylene bag, and frozen at −(18 ± 1) °C until the physicochemical analyses were performed.

All of the above steps can be shown in Figure 2.

### 2.5. Drying Kinetics

During the HAD process, the moisture content of the drying base was calculated as Equation (1) [18].
(1)$$M_{t}=\frac{W-D}{D}$$
where *M_t_* is the dry basis moisture content of seabuckthorn berries at any time, *D* is the dry weight of samples, and *W* is the weight at any one time.

The moisture ration was calculated as Equation (2) [19].
(2)$$MR=\frac{M_{t}-M_{e}}{M_{0}-M_{e}}$$
where *M_0_* is the dry basis moisture content of seabuckthorn berries at the initial drying time, *M_e_* is the equilibrium of dry basis moisture content, *M_t_* is the dry basis moisture content at any time. The value of *M_e_* is very small (that is, numbers close to zero) compared to *M_0_* and *M_t_*, thus, it can be ignored.

The drying rate was calculated as Equation (3) [20].
(3)$$DR=\frac{M_{t}-M_{t+\Delta t}}{\Delta t}$$
where Δ*t* is the time difference between a start (t) and a stop; *M_t_*, *M_t+_*_Δ*t*_ are the dry basis moisture content at time *t* and *t +* Δ*t*.

### 2.6. Color Assessment

The samples were directly transferred to HAD after different pretreatment, then ground to a fine powder by a disintegrator separately (FW 100, TAISITE Instrument, Beijing, China). The color of the powder was measured using a photoelectric colorimeter (CR 400, Konica Minolta, Tokyo, Japan). The colors of the powder were indicated by *L** (lightness)*, a** (redness/greenness) and *b** (yellowness/blueness) values. An average of six readings was taken for every same treatment. The parameters of fresh were recorded as: *L_0_** = 53.55, *a_0_** = 25.53, *b_0_** = 47.50. The differences between fresh and dried samples in color were represented by total color difference (Δ*E*), which was computed using Equation (4) [21].
(4)$$\Delta E=\sqrt{{\left(L^{*}-L_{0}^{*}\right)}^{2}+{\left(a^{*}-a_{0}^{*}\right)}^{2}+{\left(b^{*}-b_{0}^{*}\right)}^{2}}$$

### 2.7. Vitamin C Content

Preparation of standard solutions: weight of L (+)-ascorbic acid standard 100.00 mg, dissolved with 2% oxalic acid and diluted to a brown volumetric flask of 100 mL and shaken well, then a standard solution with a concentration of 1.0 mg/mL was obtained.

Extraction of vitamin C from samples: weight of the dried powder 1.00 g, added 2% oxalic acid according to solid-liquid ratio 1:20 (g/mL), extracted by ultrasonic ice bath at 200 W for 30 min, and finally centrifuged in a centrifuge (LC-LX-H185C, Lichen Bonsi Instrument Technology Co., Ltd., Shanghai, China) for 30 min at 8000 r/min, then collected the supernatant.

To avoid the effect of sample color, the vitamin C content was measured by 2,6-dichloroindophenol reverse titration method [22] and expressed on a dry basis. The content was calculated according to Equation (5).
(5)$$A=\frac{c_{1}\cdot V_{1}\cdot V_{2}}{V_{3}\cdot W}\cdot 100$$
where *A* is the content of vitamin C (mg/100 g), *c_1_* is the concentration of the standard solution, mg/mL, *V_1_* is the volume of the standard solution that was consumed for titration of 2,6-dichloroindophenol sodium salt (5 mL), mL; *V_2_* is the total volume of the supernatant, *V_3_* is the volume of the supernatant that was consumed for titration of 2,6-dichloroindophenol sodium salt (5 mL); *W* is the dry weight of the samples (g).

### 2.8. Total Flavonoid Content (TFC)

The extraction of total flavonoids from samples and the procedure was as follows: the dried powder (1.50 g) was placed in a 50 mL test tube, and 95% ethanol was added according to the solid–liquid ratio of 1:25 (g/mL), then extracted for 35 min at 200 W and 80 °C in the ultrasonic instrument, cooled to room temperature and centrifuged for 10 min (8000 r/min) [23]. Finally, we collected the supernatant. Preparation of standard solutions: rutin standard (100.00 mg) was mixed with 60% ethanol and diluted to a volumetric flask of 100 mL, then a standard solution with a concentration of 0.2 mg/mL was obtained. Finally, the standard solution was diluted to 40, 80, 160, and 320 µg/mL, respectively.

The TFC of the extraction solution was determined using a modified colorimetric method [24]. The extraction solution (1.5 mL) and NaNO2 solution (5%, 1.5 mL) were placed in a 10 mL test tube. Five minutes later, we added aluminum trichloride solution (10%, 0.4 mL) to the mixture and allowed it to stand for 5 min before NaOH solution (1 mol/L, 1 mL) and distilled water were added. The absorbance was determined at 510 nm using an ultraviolet spectrophotometer (UV-1900i, Shimadzu Co., Ltd. (Suzhou), Suzhou, China). Each determination was performed in triplicate. The TFC was expressed as mg of rutin equivalents (RE) per g of dry weight (mg RE/g, d.b.).

### 2.9. Total Phenolic Content (TPC)

Preparation of standard solutions: gallic acid standard (10.00 mg) was mixed with 60% ethanol and diluted to a volumetric flask of 10 mL, then, a standard solution with a concentration of 1.0 mg/mL was obtained. Finally, the standard solution was diluted to 20, 40, 60, 80, 100 and 320 µg/mL, respectively.

The extraction method of TPC was the same as TFC. The TPC of extracts was determined by the Folin–Ciocalteu method [25]. Extraction solution (0.4 mL) and Folin-Ciocalteu reagent solution (10%, 2 mL) were placed in a 10 mL test tube, then we added sodium carbonate solution (10%, 3 mL) to the mixture and allowed it to stand for 2 h in darkness. The absorbance was determined at 765 nm using the ultraviolet spectrophotometer. Each determination was performed in triplicate. The TPC was expressed as mg of gallic acid equivalents (GAE) per g of dry weight (mg GAE/g, d.b.).

### 2.10. Rehydration Ratio (RR)

The RR of dried samples was determined using a modified method [26]. Dried samples (2.0 g) were placed in a beaker with distilled water (100 mL) at 30 °C for 2 h. After rehydration, the samples were wiped with tissues and then weighed. The Electro-Thermostatic Water Bath was adjusted to 30 °C an hour earlier to achieve steady-state conditions. Each determination was performed in triplicate. The *RR* was calculated according to Equation (6).
(6)$$RR=\frac{M_{f}}{M_{g}}$$
where *M_f_* is the weight of dried samples was rehydrated for 2 h, *M_g_* is the initial weight of dried samples, *g*.

### 2.11. Comprehensive Evaluation

The coefficient of variation method was used to evaluate the quality attributes of dried samples after different pretreatment methods. The relevant calculations were performed according to the following equation.
(7)$$V_{i}=\frac{\sigma_{i}}{X_{i}}$$
(8)$$W_{i}=\frac{V_{i}}{\sum_{i=1}^{n}V_{i}}$$
(9)$$Z_{ij}=\frac{X_{ij}-X_{i}}{\sigma_{i}}$$
where *V_i_* is the coefficient of variation of the *i*-th indicator; *σ_i_*, the standard deviation of the *i*-th indicator; *X_i_*, mean of the *i*-th indicator; *W_i_*, the weight of the *i*-th indicator; *Z_ij_*, standardized variable value of each indicator; *X_ij_*, the measurement values of each indicator.

The comprehensive score is the sum of the scores for each indicator. The comprehensive scores of dried samples under different pretreatment methods were obtained and the steps were as follows: we multiplied the standardized variable values computed by Equation (9) and the weights calculated via Equation (8) correspondingly, then added up all [27]. We gave it a minus sign after normalization when the value showed a significant negative correlation with the indicator. The comprehensive score was calculated as Equation (10).
(10)$$S_{k}=\sum_{i=1}^{n}W_{i}\cdot Z_{ik}$$
where *S_k_* is the comprehensive score of the *k*-th pretreatment method; *Z_ik_* is the *i*-th standardized variable value of the *k*-th pretreatment method.

### 2.12. Statistical Analysis

In this study, all values are expressed as mean ± standard deviation. The experimental data were analyzed by one-way ANOVA analysis and Duncan’s multiple range tests using SPSS version 21.0 (IBM Corp., Armonk, NY, USA). The difference of statistical significance was tested at 5% probability level (*p* < 0.05). The data standardization and the weight calculation were performed using SPSS 21.0. OriginPro 2021 (Version 9.8, OriginLab Corp., Northampton, MA, USA) was used to make the chart.

## 3. Results and Discussion

### 3.1. Effects of Different Pretreatment Methods on the Surface Microstructure

The surface microstructure under different pretreatment methods is shown in Figure 3. It shows that there are some differences between the different pretreatment methods. Figure 3a shows the microstructure of the untreated seabuckthorn berry, which is free from any cracks or holes. A similar phenomenon was observed in grapes [28]. Figure 3b–d show the surface microstructure changed by the UA treatment. With the ultrasonic time increased from 5 min to 15 min, the wax layer dissolved gradually, and the micropores appeared at 15 min. As shown in Figure 3b, the epidermis had a large bubble; meanwhile, the stoma was destroyed. Figure 3e–f show that the microstructure is the same as the untreated one that has no cracks and is microporous. The results showed that PH pretreatment could produce mechanical damage to the skin of samples. Figure 3b+e–d+g are the surface microstructures of samples treated with UA + PH. It was observed that ultrasonic time affected the microstructure of seabuckthorn berry. The longer the ultrasonic time, the more the dissolution of stomata [29]. After UA15 + PH pretreatment, the stoma was destroyed and only the skeleton was visible. To sum up, once treated with UA, the wax layer would dissolve and the micropores were produced because of the alkali, mechanical and cavitation effects of ultrasonic, which could reduce the migration resistance of free water and shorten the drying time.

### 3.2. Effects of Different Pretreatment Methods on Drying Characteristics

The effects of different pretreatments on the drying kinetics of seabuckthorn berries are shown in Figure 4. With the increase in drying time, the moisture ratio decreased gradually. The results of statistical analysis revealed that the pretreated samples dried faster than the untreated ones, the UA + PH pretreatment significantly (*p* < 0.05) decreased drying time. Compared with the control group (23.6 ± 0.17 h), the drying time of PH1, PH3 and PH6 was shortened by 2.33%, 16.31% and 21.82%, respectively. This is because the wax layer is the main obstacle to moisture transfer during the drying process [30]. The treatment of PH reduced the resistance by directly destroying the skin of seabuckthorn berries, and the more micro-holes, the lower the drying time. A similar result was observed by Zhang et al. [10], who found that the drilling hole pretreatment could significantly decrease the drying time of line pepper. Compared with the control group, the drying times of UA5, UA10 and UA15 were shortened by 5.08%, 3.06% and 6.14%, respectively. This phenomenon was probably due to the treatment of UA not only accelerating the dissolution of the waxy layer but also facilitating the drying process by forming microporous in the skin through the mechanical effect of ultrasound [29]. However, the experimental results showed that the effect of ultrasonic time was not significant (*p* > 0.05), and the results were similar to those of Wan et al. [9]. In addition, the drying time of the combination pretreatment of UA + PH was shorter than that of UA and PH, where UA + PH6 was significantly (*p* < 0.05) reduced by 30.30% to 33.05% compared to the control group.

Figure 5 shows the drying rate curves of seabuckthorn berries under different pretreatment methods. The drying process of seabuckthorn berries included two stages: the early stage (t < 2 h) of drying was in a decelerated stage, and after that, the drying rate changed less and was regarded as a constant rate drying stage. This might be due to the high moisture content (w.b.) of seabuckthorn berries at the early stage of drying, and the resistance of moisture evaporation from the surface was small. As the drying continues, the free water content decreased, and the fruit oil and a small amount of nutrients gradually adhered to the tail and other micropores of seabuckthorn berries, which increased the resistance to the outward migration of moisture; at the same time, the increase in the ratio of bound water and fruit oil, the effect of binding on free water was enhanced, which affected the water evaporation process [31]. In addition, the phenomenon of ‘crusting’ and ‘wrinkling’ in the late stage of hot air drying of seabuckthorn would also increase the resistance of water diffusion and decrease the drying rate [32]. Figure 5a–c show that pretreatment mainly affects the initial drying period. The initial drying rate of the control group was 0.86 g/(g·h). The initial drying rates of UA5, UA10 and UA15 were 0.71, 0.66, and 0.67 g/(g·h), respectively. This might be due to the prolonged ultrasound time causing the damage of water migration channels on the surface of samples, which affected the mass transfer process and led to a lower rate. As shown in images (b) and (c), PH6 and UA15 + PH6 had the highest initial drying rates, they were 1.41 and 1.40 g/(g·h), respectively, indicating that increasing the quantity of holes had a significant positive effect on improving the initial drying rate.

### 3.3. Effects of Different Pretreatment Methods on Color Parameters

The color of dried products is an important organoleptic indicator that can directly affect the consumer’s choice, and the lower color difference values are more popular with consumers. The color properties of fresh and dried seabuckthorn berries are listed in Table 1. The *L**, *a** and *b** values of dried seabuckthorn berries were lower than those of the fresh ones, but compared with the control group, the *L** value increased. The results indicated that the pretreatment methods of PH and UA could effectively maintain the original brightness and make it closer to the fresh fruit. The *L** value of UA15 + PH6 samples was closest to that of the fresh fruit, and the control group had the farthest value. The fresh fruit had the highest *a** value, followed by UA5, and the lowest for UA15 + PH1. Similarly, the highest *b** value was observed for UA15 + PH6, except for the fresh samples, and the lowest for CG. The UA15 + PH6 samples had the lowest Δ*E* value. Specifically, the samples that were pretreated with UA had a smaller color difference when compared to control samples, and the shorter the treatment time, the smaller the color difference. This can be explained by the fact that the yellow pigment loss was due to the excessive dissolution of the wax layer of samples, which affected the overall color [33]. The Δ*E* value of PH-treated samples decreased first and then increased, among which, PH3 was the smallest. It is probably due to the shorter drying time and the less browning reaction. However, as the hole number of PH pretreatment increased, the loss of juice and the yellow pigment increased, the *b** value decreased and the Δ*E* value increased. Among all experimental groups, dried samples pretreated for UA15 + PH6 had the highest *L**, *b** values and the lowest Δ*E* value.

### 3.4. The Content of Vitamin C

The effects of different pretreatment methods on the vitamin C of seabuckthorn berries are shown in Figure 6. Vitamin C is highly instable and easy to decompose under light, high temperature and oxygen [34]. Samples pretreated with PH had higher vitamin C content than those pretreated with UA and UA + PH. Moreover, the highest value of vitamin C content was shown in PH3 samples (54.71 mg/100 g), followed by control samples (53.94 mg/100 g). Those values were significantly (*p* < 0.05) higher than that of the other pretreatment groups. This is because the samples from the PH group and the control group were not subjected to complex pretreatment steps and chemical effects of lye, and the PH3 samples had a short drying time and less juice loss [14]. The vitamin C contents of UA and PH samples were increased first, and then decreased with the rising of ultrasonic treatment time and number of the perforations, respectively. Of all the samples pretreated with UA + PH, the UA10 + PH3 samples had the highest value of vitamin C content, the UA15 + PH1 samples had the lowest value of vitamin C content. This might be due to the seabuckthorn berries being affected by ultrasound and lye for a long time, causing the pH level to increase, the waxy layer to be overly dissolved and the juice leakage, which negatively affected the retention of vitamin C [35].

### 3.5. Total Flavonoid Content (TFC)

Figure 7 shows the TFC of the samples that were treated with different pretreatment methods. Compared with the control group, the TFC of samples from UA5, UA10 and UA15 increased by 14.56%, 12.30% and 6.88%, respectively. Thus, the pretreatment method of UA had a positive effect on the increase in TFC [36], and the shorter the ultrasound time, the better the results. This is probably because the pretreatment of UA helped rupture the fruit cell walls and promote the release of flavonoid compounds from the matrix of seabuckthorn berries, resulting in the increase in TFC. However, the excessive accumulation of free radicals would lead to the degradation of flavonoids due to extended treatments [37]. The TFC of the PH1 samples was higher than that of the PH6 samples, which was mainly because the appropriate number of holes could not only reduce the drying time but also prevent the great loss of flavonoids in the drying process. In the combined pretreatment groups, the TFC of UA5 + PH samples was higher than the control and the PH samples but lower than the UA5 group. This is because when using the UA5 + PH pretreatment method, the positive effect of ultrasound can offset the loss caused by PH treatment.

### 3.6. Total Phenolic Content (TPC)

Total phenolic in seabuckthorn berries is an important bioactive compound. It has a wide range of pharmacological activities, including anti-cancer, anti-aging, anti-inflammatory, etc. [38]. Meanwhile, phenolics are sensitive to heat, oxygen and alkali [39]. The effects of different pretreatment methods on the TPC of seabuckthorn berries are shown in Figure 8. The TPC of samples treated with UA5 was higher than untreated ones (12.53 mg/g), possibly because the ultrasonic action helped promote the release of phenolic compounds and reduce the drying time, which has a positive effect on the retention of TPC. However, the values of TPC decreased with the increase in UA treatment time, the highest value (13.40 mg/g) was observed in the drying of samples treated for UA5. This can be attributed to the shorter drying time, and less oxidation and thermal decomposition of phenolics [40,41]. On the other hand, the shorter treatment time, the lower degree of wax layer dissolution and less juice loss than UA15. Among all PH groups, the highest and the lowest values were presented at PH1 and PH6, respectively. It is probably due to the effect of juice leakage [40]. The juice of seabuckthorn berries has plenty of phenolic compounds [42]. For the UA + PH groups, the samples of UA5 + PH3 had the highest TPC value, which was higher than UA5 and PH3. The higher TPC is due to the release of phenolic substances caused by the pretreatment of UA. Similarly, the shorter drying time caused by the pretreatment of PH could reduce the phenolics loss.

### 3.7. Rehydration Capacity of Seabuckthorn Berries

Rehydration capacity is an important quality index for evaluating the damage to the cellular structure of dried samples [43]. The effects of different pretreatment methods on rehydration ratio (RR) are illustrated in Figure 9. It follows that the pretreatment method of UA had no significant (*p* > 0.05) influence on the RR compared with the control, and the RR for UA samples was lower than the control (1.07 *g/g*). The reason is probably that the cellular structure was damaged by the combined effect of ultrasound and alkali [44]. Samples treated with PH had stronger rehydration capacity than untreated samples, and the higher the number of holes, the higher the value of RR. A similar result was observed by Chen et al. [11], who reported that blueberries treated with CO_2_ laser perforation had a higher RR due to the formation of holes in the skin. The results indicated that the porous structure helped promote moisture filling, and the shorter drying time could reduce the microstructure damage caused by thermal energy [45]. The samples from the group of UA + PH had a higher RR than the UA group, and the samples of UA15 + PH6 had the highest RR (1.15 *g/g*), which could be attributed to the shorter drying time.

### 3.8. Comprehensive Score

The coefficient of variation method is a weighting way to calculate the weight according to the information contained in each evaluation index. It can avoid the subjectivity of artificial assignment and reflect the index relative important degree objectively [46]. This method was based on the known mean and standard deviation, and through the corresponding formula to determine the coefficient of variation and weight of each indicator from dried samples. As shown in Table 2, the biggest weight appeared in “drying time” (20.5%), followed by “vitamin C content” (18.2%), and then the “*b**” (16.0%). The results showed that different pretreatment methods had a great influence on these three indicators, through which the appropriate pretreatment methods can be quickly judged.

Before the comprehensive evaluation, we should use “z-score” to standardize the values for removing the influence of dimension and ensuring the comparability between indicators. The standardized values of each indicator and the comprehensive scores of different pretreatment methods are shown in Table 3. The samples treated with UA15 + PH6 had the highest comprehensive score (1.013). In other words, the pretreatment method of UA15 + PH6 is very suitable for seabuckthorn berries [47].

## 4. Conclusions

In this study, we investigated the drying characteristics and quality attributes of seabuckthorn berries dried by HAD with UA, PH, UA + PH as pretreatment methods. The microstructures showed that the wax layer was dissolved and the micropores were produced by lye, mechanical and cavitation effects of ultrasonic. The HAD process of seabuckthorn berries consisted of a reduced-speed drying, followed by a constant-speed drying. The UA and PH pretreatment had a synergistic effect on drying characteristics. The samples treated with UA15 + PH6 had the shortest drying time (15.8 h) and the highest drying rate (1.40 g/(g·h)). The best values of Δ*E* (29.61) and rehydration ratio (1.15 *g/g*) were found in the group UA15 + PH6. The highest value of vitamin C content appeared in the samples of PH3 (54.71 mg/100 g). The UA5 and PH1 samples had the highest value of total flavonoid content (11.41 mg/g) and total phenolic content (14.20 mg/g), respectively. Finally, the coefficient of variation method was used to calculate the comprehensive scores of samples that from different groups, and the highest score (1.013) was observed in the group of UA15 + PH6. From these results, it can be concluded that UA15 + PH6 treatment was the most appropriate pretreatment method for seabuckthorn berries. The next step of research could be on how to reduce the juice loss during the UA + PH pretreatment.

## Figures and Tables

**Figure 1 foods-11-03675-f001:**
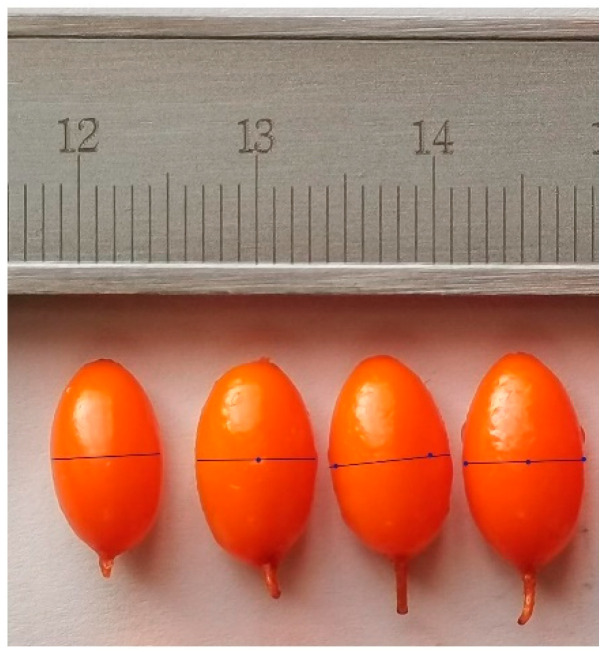
Schematic diagram of the pretreated location. Line indicates the place of the maximum diameter; dot indicates the location of the perforation.

**Figure 2 foods-11-03675-f002:**
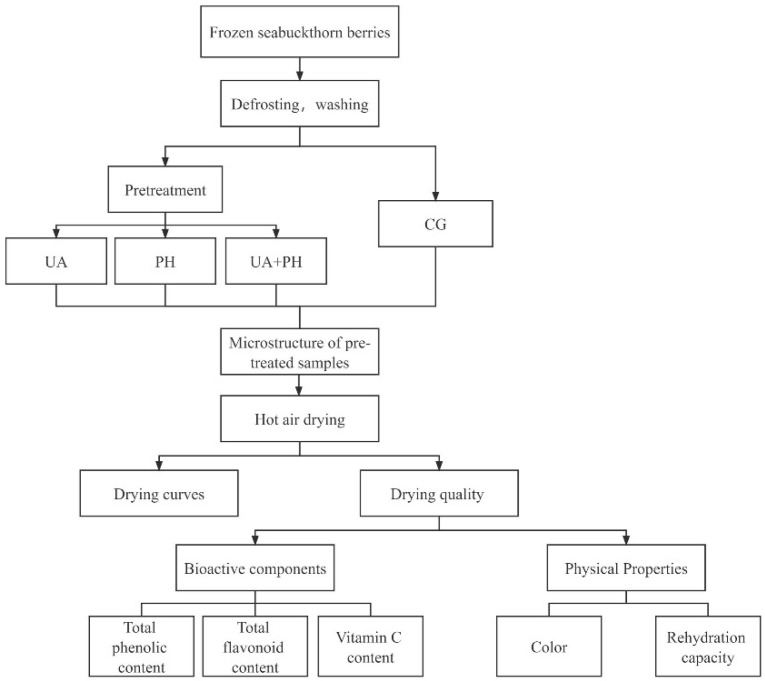
Diagram of the whole experimental procedure.

**Figure 3 foods-11-03675-f003:**
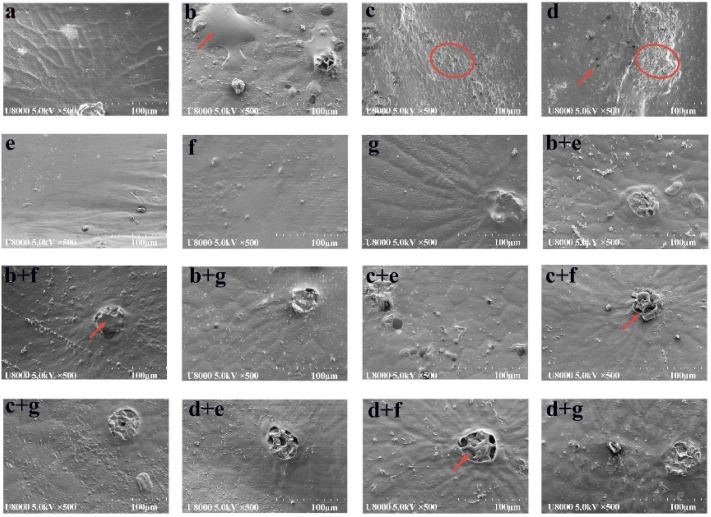
Effects of different pretreatment methods on surface microstructure of dried samples: (**a**) CG; (**b**) UA5; (**c**) UA10; (**d**) UA15; (**e**) PH1; (**f**) PH3; (**g**) PH6; (**b+e**) UA5 + PH1; (**b+f**) UA5 + PH3; (**b+g**) UA5 + PH6; (**c+e**) UA10 + PH1; (**c+f**) UA10 + PH3; (**c+g**) UA10 + PH6; (**d+e**) UA15 + PH1; (**d+f**) UA15 + PH3; (**d+g**) UA15 + PH6.

**Figure 4 foods-11-03675-f004:**
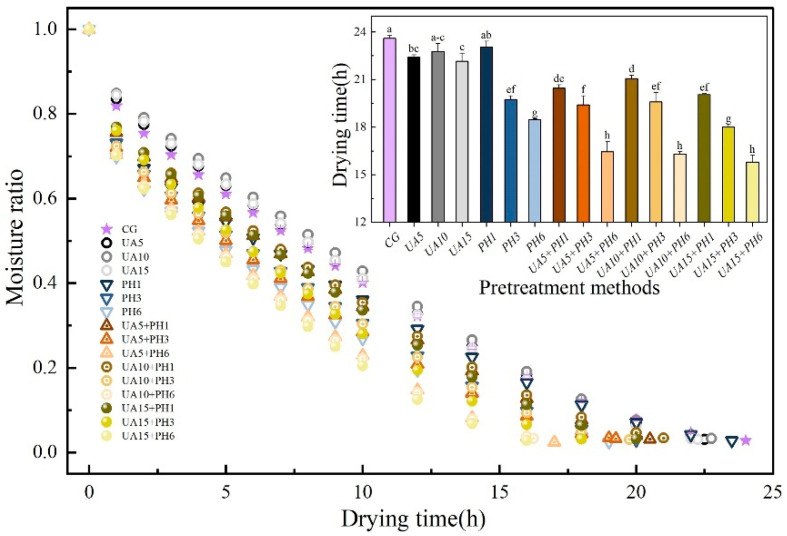
Effects of different pretreatment methods on drying kinetics of seabuckthorn berries dried at 60 °C and 2.2 m/s. Different lowercase letters indicate significant differences between samples (*p* < 0.05). (If there are more than two, use the abbreviated form. For example, abc is written a–c).

**Figure 5 foods-11-03675-f005:**
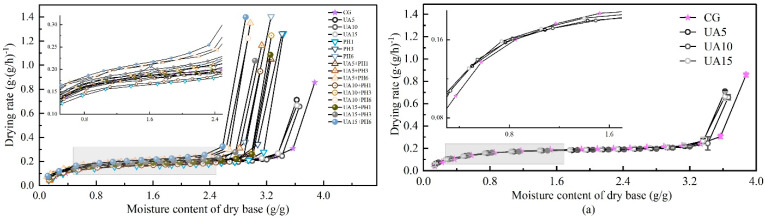
Effects of different pretreatment methods on drying rate of seabuckthorn berries dried at 60 °C and 2.2 m/s: (**a**) Drying rate of UA treated; (**b**) Drying rate of PH treated; (**c**) Drying rate of UA + PH treated.

**Figure 6 foods-11-03675-f006:**
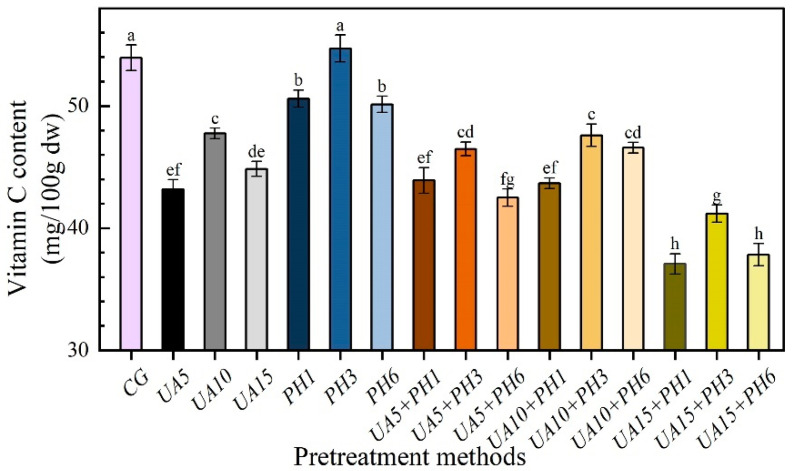
Effects of different pretreatment methods on Vitamin C content of seabuckthorn berries. Notes: Different lowercase letters indicate significant differences between samples (*p* < 0.05). (If there are more than two, use the abbreviated form. For example, abc is written a–c).

**Figure 7 foods-11-03675-f007:**
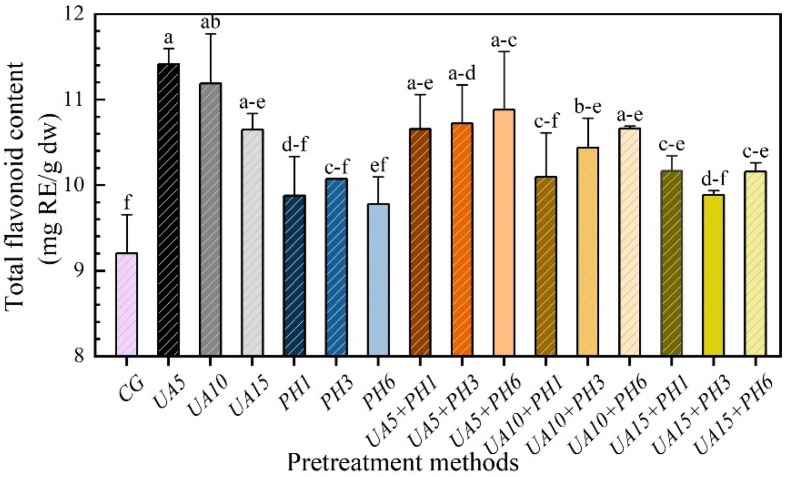
Effects of different pretreatment methods on TFC of seabuckthorn berries. Notes: Different lowercase letters indicate significant differences between samples (*p* < 0.05). (If there are more than two, use the abbreviated form. For example, abc is written a–c).

**Figure 8 foods-11-03675-f008:**
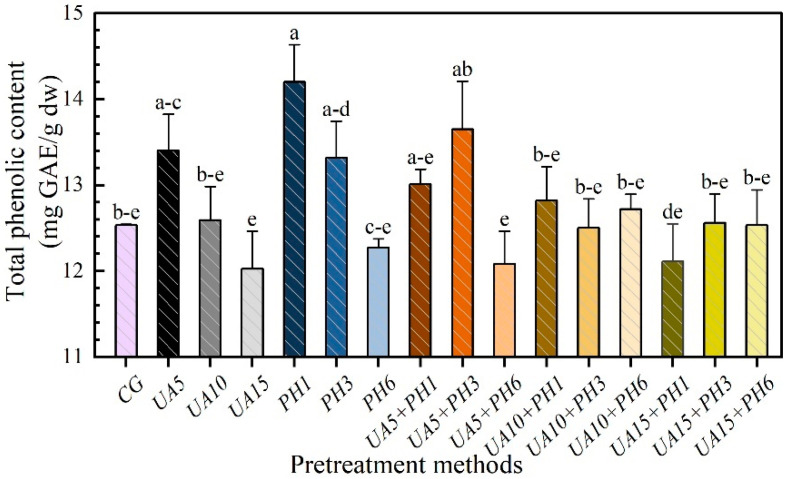
Effects of different pretreatment methods on TPC of seabuckthorn berries. Notes: Different lowercase letters indicate significant differences between samples (*p* < 0.05). (If there are more than two, use the abbreviated form. For example, abc is written a–c).

**Figure 9 foods-11-03675-f009:**
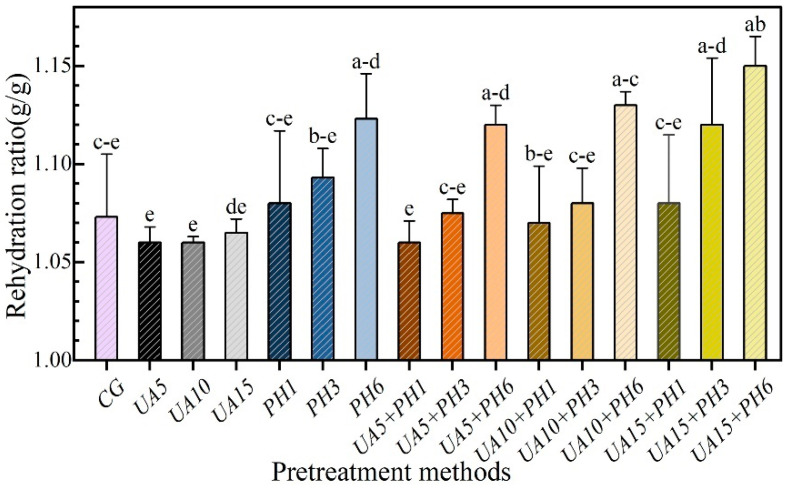
Effects of pretreatment on rehydration ratio of seabuckthorn berries. Notes: Different lowercase letters indicate significant differences between samples (*p* < 0.05). (If there are more than two, use the abbreviated form. For example, abc is written a–c).

**Table 1 foods-11-03675-t001:** Effects of different pretreatment methods on the color parameters of seabuckthorn berries.

Pretreatment Methods	*L**	*a**	*b**	Δ*E*
Fresh 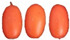	53.54 ± 0.25 ^a^	25.55 ± 0.30 ^a^	47.51 ± 0.22 ^a^	--
CG 	28.37 ± 0.48 ^j^	17.32 ± 0.32 ^fg^	19.71 ± 0.7 ^h^	38.4 ± 0.79 ^a^
UA5 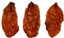	30.87 ± 0.27 ^d^	19.27 ± 0.36 ^b^	23.82 ± 0.29 ^d^	33.39 ± 0.35 ^h^
UA10 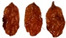	30.48 ± 0.42 ^de^	18.62 ± 0.21 ^d^	23.08 ± 0.37 ^de^	34.3 ± 0.54 ^gh^
UA15 	29.63 ± 0.45 ^fg^	17.76 ± 0.25 ^f^	21.45 ± 0.36 ^g–i^	36.21 ± 0.59 ^c–f^
PH1 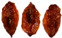	28.83 ± 0.48 ^h–j^	17.4 ± 0.19 ^fg^	20.72 ± 0.35 ^ij^	37.34 ± 0.57 ^a–c^
PH3 	29.87 ± 0.29 ^ef^	18.34 ± 0.29 ^f^	22.29 ± 0.23 ^e–g^	35.33 ± 0.39 ^fg^
PH6 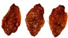	29.19 ± 0.34 ^f–i^	17.44 ± 0.32 ^ef^	21.11 ± 0.21 ^h–j^	36.82 ± 0.31 ^b–e^
UA5 + PH1 	29.89 ± 0.21 ^ef^	17.72 ± 0.2 ^f^	22.47 ± 0.28 ^ef^	35.32 ± 0.32 ^fg^
UA5 + PH3 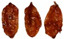	29.41 ± 0.27 ^f–h^	17.88 ± 0.49 ^ef^	21.95 ± 0.55 ^f–h^	35.98 ± 0.65 ^d–f^
UA5 + PH6 	29.59 ± 0.4 ^fg^	17.58 ± 0.27 ^fg^	22.29 ± 0.43 ^e–g^	35.69 ± 0.58 ^ef^
UA10 + PH1 	29.19 ± 0.44 ^f–i^	17.5 ± 0.2 ^fg^	20.79 ± 0.32 ^ij^	37.03 ± 0.48 ^b–d^
UA10 + PH3 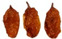	31.58 ± 0.48 ^c^	18.8 ± 0.31 ^cd^	26.1 ± 0.53 ^c^	31.4 ± 0.74 ^i^
UA10 + PH6 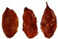	28.95 ± 0.33 ^g–j^	17.01 ± 0.39 ^gh^	20.42 ± 0.5 ^jh^	37.57 ± 0.65 ^ab^
UA15 + PH1 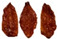	28.56 ± 0.24 ^ij^	16.71 ± 0.35 ^h^	19.72 ± 0.5 ^h^	38.4 ± 0.57 ^a^
UA15 + PH3 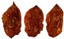	29.07 ± 0.33 ^g–i^	17.63 ± 0.24 ^f^	21.41 ± 0.43 ^g–i^	36.64 ± 0.49 ^b–e^
UA15 + PH6 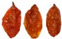	32.92 ± 1.29 ^b^	18.36 ± 1.04 ^de^	27.52 ± 2.01 ^b^	29.61 ± 2.48 ^j^

Data are expressed as mean ± standard deviation. Different lowercase letters in the same column indicate that the mean values are significantly different (*p* < 0.05). Symbols: *L******—lightness; *a**—redness/greenness; *b**—yellowness/blueness; Δ*E*-color difference.

**Table 2 foods-11-03675-t002:** Data of each indicator under the coefficient of variation method.

Indicator	Mean	Standard Deviation	Coefficient of Variation	Weight (%)
Drying time	19.953	2.482	0.124	20.5
*L**	29.774	1.191	0.040	6.6
*a**	17.833	0.680	0.038	6.3
*b**	22.178	2.147	0.097	16.0
Δ*E*	35.588	2.422	0.068	11.2
vitamin C content	0.457	0.05	0.110	18.2
TFC	10.363	0.567	0.055	9.0
TPC	12.770	0.608	0.048	7.8
RR	1.090	0.029	0.027	4.4

**Table 3 foods-11-03675-t003:** Comprehensive score of dried seabuckthorn under different pretreatment methods.

Pretreatment Method	Drying Time	*L**	*a**	*b**	Δ*E*	Vitamin C Content	TFC	TPC	RR	Comprehensive Score	Rank
CG	0.311	−0.081	−0.049	−0.19	0.134	0.305	−0.191	−0.032	−0.026	−0.709	15
UA5	0.209	0.063	0.137	0.126	−0.105	−0.096	0.172	0.084	−0.047	0.335	4
UA10	0.239	0.04	0.075	0.069	−0.062	0.074	0.135	−0.024	−0.047	0.145	7
UA15	0.187	−0.008	−0.007	−0.056	0.03	−0.034	0.046	−0.099	−0.039	−0.414	13
PH1	0.264	−0.054	−0.042	−0.112	0.084	0.18	−0.08	0.19	−0.016	−0.282	12
PH3	−0.017	0.005	0.048	0.008	−0.012	0.333	−0.048	0.073	0.005	0.453	3
PH6	−0.128	−0.033	−0.038	−0.082	0.059	0.163	−0.097	−0.067	0.052	−0.033	9
UA5 + PH1	0.042	0.007	−0.011	0.023	−0.013	−0.068	0.048	0.032	−0.047	−0.045	10
UA5 + PH3	−0.047	−0.021	0.004	−0.017	0.019	0.027	0.059	0.117	−0.023	0.174	6
UA5 + PH6	−0.299	−0.011	−0.024	0.008	0.005	−0.121	0.085	−0.092	0.047	0.186	5
UA10 + PH1	0.094	−0.033	−0.032	−0.107	0.069	−0.077	−0.044	0.007	−0.031	−0.48	14
UA10 + PH3	−0.03	0.103	0.092	0.301	−0.2	0.069	0.012	−0.036	−0.016	0.755	2
UA10 + PH6	−0.312	−0.047	−0.078	−0.135	0.095	0.031	0.049	−0.007	0.063	0.093	8
UA15 + PH1	0.008	−0.069	−0.107	−0.189	0.134	−0.322	−0.033	−0.088	−0.016	−0.966	16
UA15 + PH3	−0.167	−0.04	−0.02	−0.059	0.05	−0.169	−0.079	−0.028	0.047	−0.231	11
UA15 + PH6	−0.354	0.18	0.05	0.41	−0.286	−0.295	−0.034	−0.032	0.094	1.013	1

## Data Availability

The data used in this study are available in this article.

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
