# Peer review of "Hot Air Drying of Seabuckthorn (Hippophae rhamnoides L.) Berries: Effects of Different Pretreatment Methods on Drying Characteristics and Quality Attributes"

_foods, 2022, doi:10.3390/foods11223675_

Round 1

Reviewer 1 Report

Review Report for Scientific Reports Journal

There are several concerns regarding the manuscript because of which I do not think the manuscript could be accepted in a reputed journal like this in its current form. The paper requires major proofreading to improve the grammar and presentation:

Major Comments

1.      I suggest a brief discussion, perhaps after the introduction of previous studies that have utilised the same concept and techniques with an emphasis on the novelty introduced as part of this work and how gaps are being bridged.

2.      Please highlight the limitations of the study and the scope for future research based on the findings of the study.

3.      Abstract require major corrections. Some of the points I am indicating:

First line of the abstract is an incomplete sentence. “To shorten the hot air drying (HAD) time of seabuckthorn berries, improve the quality of 10 dried products.” It does not make any sense.

Line 11-14: too much information in one line. Consider splitting it.

Line17: drying efficiency. How did you measure that? Nothing in methods and results about that.

Line 20: All the parameters could. Is that your result, or you are predicting? I think it is your result, then why you are using the word could.

Line 24: highest quality comprehensive score. Mention the value.

Line 25: it is the best for cherries and would be best for others. Looking at your sentence, it looks like you are making some sort of hypothesis.

4.      Introduction:

Line 33: reference for the dimension and weights. Please add a reference in every place where you use numerical values from the literature.

Line 36: rich in nutrients. Add values if possible.

Line 38: 70-85%. Add a reference for that.

Line 44: at present. add reference

Line 48: in conclusion. That is quite a wrong way to write this type of conclusive word in the introduction.

Line 59: significantly shorten. Add values with reference

Line 61-63: More critical details are required on the research gap.

5.      Materials and method:

Line 73-75: is it for your current study or from literature? If it is from your study, then why do you put reference?

Mean weight and size of the samples should be reported in the method.

Line 95: A diagram of the perforated sample would help to understand the reader in a more appropriate manner.

Line 99-117: At least one picture of the samples from each group is required to make it reader-friendly.

Section 2.3: I am lost here. Is it a mixed mode of drying? If you only did HAD, then why freeze-drying as well?

Line 127-129: So, it is not a fabricated dryer; instead a dryer from the shop.

Line 129-131: consider make it one sentence.

Line 135: Instead of that “The drying test was 135 stopped when the final wet basis moisture content was lower than 15% (w.b.) or dry basis 136 (d.b.) moisture content was lower than 0.18 g/g [16]” consider writing that “The drying test was 135 stopped when the final wet basis moisture content was lower than 15% (w.b.) or dry basis 136 (d.b.) moisture content was lower than 0.18 g/g following the method mention in [16]”

Figure 1: Microstructure: write “microstructure of pre-treated samples”

Consider showing the microstructure of the dried samples as well.

Section 2.12: more details are required

Line 228-231: Try to express them as an equation if possible. This is the main conclusive result of the whole study based on which you have recommended the final method. Therefore, more clarification on comprehensive scores is required.

There are many other aspects which are quite painstaking for the reviewer. In short, I would highly recommend proofreading the whole draft from a native speaker.

Therefore, author needs to undergo all these remarks.

Reviewer 2 Report

Line 10-13, 346-347: Revised the sentences.  

Line 99: Control group (CK)? Looks confusing.

Figure 1: Revised the figure and provide a clearer version.

The author should use the updated template of the journal.

Figure 4: The author should update the figures with clear images.

Figure 8: Some of the data standard errors look very high. That put a question mark on the experimental process and data collection. The author should check the data analysis and review/justify that the experimental process was adopted right.

The author should mention all the numerical values of the quality parameters obtained from best processing in the Abstract and conclusion sections.  

The manuscript needs English revision to address sentence complexity and increase the coherent of the study

Reviewer 3 Report

Foods-2001324

The manuscript presents the results of hot air drying of Seabuckthorn berries and the effects of specific pretreatments on some quality attributes. In general, the document presents interesting results for this product. Some comments that the authors must attend to clarify the methodologies and the presentation and analysis of their results:

Authors must review English throughout the manuscript. For example, in the title, the word Quality is misspelled; these errors should be reviewed and corrected throughout the document.

Line 46 – revise, the sentence is not entirely understood, what you want to highlight with …and the quality of dried cannot be guaranteed...

Lines 50-51- review which you want to highlight with ... infection. The wax layer has a very good hydrophobic which prevents ...

line 59 - review, "2% grape drier"

line 66- review and clarify, in the objective, the authors mention "select the most suitable pretreatment method..." but do not indicate why?

Lines 71-74 - it is unclear if the berries were frozen. The authors must clarify what they did before the moisture content determination (thawed, ground, etc.)

Lines 76-77- it is unclear if the berries were kept frozen or refrigerated

lines 228-231 - the authors should provide more information on this methodology and include some references. Also, clarify how the data was normalized and the weight given to the chosen responses.

In the caption of the figures that contain boxes, it must be clarified what information is presented.

Line 419 - this section should be reviewed and clarify the analysis performed. Likewise, explain how the values presented in Tables 2 and 3 were calculated. Also, include a deeper discussion on this approach and include references.

In the conclusions, line 436, revise what you mean by seed drying.

Lines 440-441 - review. It is not fully understood what is to be highlighted: “The highest value of vitamin C content appeared in the samples 440 of PH3 due to shorter drying time and less juice loss and no alkali effects.”

line 447 – delete-  6. Patents 

 Author Response

Round 2

Reviewer 1 Report

Please fix equation no. 10 and try to increase the resolution of the figures for better representation. 

Reviewer 3 Report

The authors took into account and responded appropriately to the comments and suggestions of the reviewers. The manuscript version was improved
